# The Effect of Sea Salt, Dry Sourdough and Fermented Sugar as Sodium Chloride Replacers on Rheological Behavior of Wheat Flour Dough

**DOI:** 10.3390/foods9101465

**Published:** 2020-10-14

**Authors:** Andreea Voinea, Silviu-Gabriel Stroe, Georgiana Gabriela Codină

**Affiliations:** Faculty of Food Engineering, Stefan cel Mare University of Suceava, 720229 Suceava, Romania; andreea.musu@fia.usv.ro (A.V.); codina@fia.usv.ro (G.G.C.)

**Keywords:** sea salt, dry sourdough, fermented sugar, dough rheological properties, optimization, response surface methodology

## Abstract

The aim of this study was to investigate the effects of formulation factors, sea salt (SS), dry sourdough (SD) and fermented sugar (FS) as sodium chloride replacers in wheat flour on dough mixing, extension, pasting and fermentation rheological properties, evaluated by Farinograph, Extensograph, Amylograph and Rheofermentometer devices. With regard to mixing and extension properties, SS and FS presented a strengthening effect, whereas SD presented a weakening one. SS and FS presented a positive effect on dough stability, energy and resistance, whereas SD presented a negative one. On the Amylograph, peak viscosity increased by SS and FS addition and decreased when SD was incorporated in the dough recipe. During fermentation, dough development and gas production in the dough system increased after SS and SD addition, whereas they decreased after FS addition. Response surface methodology (RSM) was used to investigate the effect of independent variables on the rheological properties of the dough. Mathematical models between the independent variables, SS, SD and FS, and the dependent variables, represented by the rheological values of the dough, were obtained. The best formulation obtained was of 0.30 g/100 g SS, 0.50 g/100 g SD and 1.02 mL/100 g FS addition with a 0.618 desirability value, following Derringer’s desirability function approach. For this formulation, bread quality characteristics were better appreciated than for those obtained for the control sample, in which 1.5% NaCl was incorporated in wheat flour.

## 1. Introduction

High salt intake is associated with high blood pressure, a major risk factor for diseases such as stroke, heart attack and cardiovascular ones [1]. The World Health Organization’s (WHO) recommendation is to reduce sodium intake to up to 2 g daily and not to exceed salt consumption of more than 5 g per day [2,3]. However, daily salt intake exceeds the WHO recommended intake in most countries. For example, in the EU, daily salt intake varies between 7 and 13 g, which largely exceeds the WHO recommendation [4]. Therefore, in many EU countries and beyond, measures are being taken in order to reduce the salt content in food products [5]. Among foodstuffs, one of the main salt sources is bakery products [6]. The salt content in bakery products may vary from country to country from around 1 to 3 g, depending on consumption habits [5]. A total replacement of sodium chloride in bakery products is very difficult to achieve due to its technological effects on the bread baking process and on the baked product quality, especially flavor and taste [7,8]. The most important impact of sodium chloride salt is on the flavor profile of the foodstuffs, which is significant due to the fact that it confers a unique salty taste [9]. Sodium chloride, even in small amounts, is one of the main ingredients in bakery products, with a major impact on the wheat flour dough rheological properties and the finished product quality. Today, many approaches are being tried to reduce the sodium content from bakery products. The most common ones are the replacement of sodium chloride with other salts, especially with chloride ones such as potassium, calcium, magnesium, etc. [10,11,12]. Lately, there has also been a concern about the use of various ingredients that may intensify the baked products’ flavor perception in order to reduce the sodium amount indifferent foodstuffs [13,14,15,16]. The aim of this study was to investigate the effect of a sodium chloride replacement in bakery products with a different salt, such as sea salt, with a low sodium content, in combination with two enhancers of baked products’ flavor, such as dry sourdough and fermented sugar, on the rheological properties of dough by using response surface methodology (RSM). Although different studies were carried out on the effects of different types of a singular formulation on dough rheological properties, such as sourdough [7,17,18] or sea salt as sodium chloride replacements [12,19], very few studies have been conducted on the combined effect of different salts, such as sodium chloride replacers [20] and different improvers of baked products’ flavor [21]. There are many approaches to reducing sodium chloride content in bakery products while maintaining their high quality. The most common ones are sodium chloride substitution with different replacers, such as magnesium salts, KCl, calcium salts, etc. [10,11,12,20], and intensification through the addition of different ingredients. This study is a continuation of previous research by our group on sodium reduction in bakery products by partial substitution of NaCl with KCl [20] or sea salt with low sodium content, in combination with dry sourdough, in order to improve the bakery products’ flavor [21]. Within this study, we used sea salt for sodium replacement in bakery recipes in combination with the other two ingredients (dry sourdough and fermented sugar) as improvers of bread flavor perception. It was taken into account that the intensification of the bakery products’ sensory characteristics, with regard to taste and flavor, is currently one of the main concerns of specialists in the field due to the fact that consumers find it difficult to accept products without the specific salty taste provided by sodium chloride. To our knowledge, this is the first study performed on the combined effect of sea salt–dry sourdough–fermented sugar formulation as sodium chloride replacers on dough rheological properties.

## 2. Materials and Methods 

### 2.1. Materials

Wheat flour provided by S.C. Mopan S.A. (Suceava, Romania) from the 2019 harvest was used. Low sodium sea salt, dry sourdough and fermented sugar were used as ingredients. Sea salt (SS) obtained from the Dead Sea was provided by BK Giulini Corp., United States, with the commercial name Salona, and presented the following content: sodium as sodium chloride (max. 7%), potassium chloride (21 ÷ 27%) and magnesium chloride (31 ÷ 35%), water insoluble max. 0.1%. Dry sourdough (SD) provided by Enzymes & Derivates S.A. Company (Neamt, Romania) was fermented wheat flour. The commercial SD product name is Grande Sélection Ble N°1, which is produced by AIT Ingredients (Soufflet Group, Saint-Maximin, France). The SD was in powder form and of a white color, with a humidity value less than 12%. According to the product technical sheet provided by the producer, it may contain yeast and molds at less than 5000 UFC/g, coliforms at less than 100 UFC/g and aerobic mesophilic flora at less than 100,000 UFC/g. The fermented sugar (FS), obtained from sugars such as beet, cane, corn and tapioca, provided by Corbion (Amsterdam, Netherlands) (product name Verdad F95), contained fermentation products, such as organic acids, residual sugars and aroma components (slightly acidic, combined with umami and bouillon notes). The product was in liquid form, which presents, according to its technical sheet, a dry matter of 54 ÷ 61%, a pH amount of 5.0 ÷ 5.6 and an amount of organic salts of 490 ÷ 630 meq/100 g. It is of a high purity with a sugar amount of 5%, arsenic, cadmium, lead and mercury amounts of maximum 1 mg/kg and heavy metals of maximum 5 mg/kg. From the microbiological point of view, it is free of *Escherichia coli, Enterobacteriaceae and Staphylococcus* in 1 g and *Salmonella* in 25 g. The *Mesophilic* bacteria is of max. 3000 counts/g and total combined yeasts and molds count (TYMC) of max. 100 counts/g. The wheat flour used was very good for bread making with low α amylase activity [22] according to the characteristics analyzed by the international standard methods: 0.65 g/100 g ash (according to the International Association for Cereal Science and Technology—ICC 104/1); 12.67 g/100 g protein (ICC 105/2); 30 g/100 g wet gluten (ICC106/1); 14.0 g/100 g moisture (ICC 110/1); falling number 442 s (ICC 107/1). 

### 2.2. Dough Rheological Properties during Mixing and Extension

During mixing, dough rheological properties were analyzed according to the ICC method 115/1 by using a Farinograph device (Brabender, Duigsburg, Germany, 300 g capacity), and during extension, according to ICC method 114/1, by using an Extensograph device (Brabender, Duigsburg, Germany). With the Farinograph, the following values were determined: water absorption (WA), dough development time (DT), dough stability (ST) and the degree of softening at 10 min (DS). With the Extensograph, the following values were determined: resistance to extension (R_50_), maximum resistance to extension (R_max_), extensibility (Ext), energy (E) and ratio number (R/E) at a proving time of 135 min.

### 2.3. Dough Rheological Properties during Pasting

The pasting properties of wheat flour dough were analyzed according to the ICC method 126/1 by using an Amylograph (Brabender OGH, Duisburg, Germany). With the Amylograph, the following values were determined: gelatinization temperature (T_g_), temperature at peak viscosity (T_max_) and peak viscosity (PV_max_).

### 2.4. Dough Rheological Properties during Fermentation

During fermentation, dough rheological properties were determined according to the American Association of Cereal Chemists (AACC) method 89-01.01 by using a Rheofermentometer device (Chopin Rheo, type F3, Villeneuve-La-Garenne Cedex, France). With the Rheofermentometer, the following values were determined: maximum height of gaseous production (H’m), total CO_2_ volume production (VT), volume of the gas retained in the dough at the end of the test (VR) and retention coefficient (CR).

### 2.5. Bread-Making Samples

Bread-making samples were made for the optimized samples (in which the optimum formula was incorporated, namely 0.30% SS, 0.50% SD and 1.02 mL FS in 100 g wheat flour) and for control samples in which 1.5% NaCl was incorporated into the bread recipe. The bread sample formula also contained 3% yeast of *Saccharomyces cerevisiae* type and deionized water, according to the water absorption value obtained with the Farinograph device. The ingredients were mixed at 200 rpm in a laboratory mixer (Kitchen Aid, Whirlpool Corporation, Benton Harbor, MI, USA) for 15 min, then divided, molded and fermented in a fermentation chamber (PL2008, Piron, Cadoneghe, Padova, Italy) at 35 °C for 40 min at 85% relative humidity and baked in a bakery convection oven (Caboto PF8004D, Cadoneghe, Padova, Italy) at 180 °C for 50 min. After cooling for 4 h, the samples were analyzed.

### 2.6. Bread Samples Analysis

The bread samples obtained through the baking tests were analyzed from the physical, color, textural and sensory points of view. The bread physical characteristics analyzed were the loaf volume, elasticity and porosity, according to the Romanian standard SR 91:2007. The bread color characteristics analyzed were *L** (lightness), *a** (redness when positive and greenness when negative) and *b** (yellowness when positive and blueness when negative) using a Konica Minolta CR-700 colorimeter (Chiyoda, Tokyo, Japan). The bread textural characteristics analyzed were springiness, cohesiveness, gumminess, firmness and resilience with a textural analyzer (Perten TVT 6700, Hägersten, Sweden). The bread sensory characteristics were analyzed by using a hedonic test of 9 points (from 1—strong dislike to 9—excellent taste) with 30 semi-trained panelists from the Stefan cel Mare University’s Faculty of Food Engineering. The bread sensory characteristics evaluated were: appearance, color, flavor, taste, smell, texture and overall acceptability. 

### 2.7. Experimental Design and Statistical Analysis

In order to achieve the proposed purpose—the analysis of the synergic effect of the amounts of sea salt (SS), dry sourdough (SD) and fermented sugar (FS) on the rheological properties of dough—the response surface method (RSM) was used. The central composite design (CCD) methodology was used. CCD methodology consists of a three-level factorial design, a start design, and a central point. In this case, for three factors (SS, SD and FS) at three levels each, the design requires nine experimental combinations which are produced with six replicates at the center point, generating a total of twenty experimental runs. The combination in the center point of the experiment is replicated six times. The factors studied were at the addition levels recommended by the producers to be incorporated in wheat flour. For SD and FS, the producers recommended levels which are between 0.5–5 g/100 g (SD) between 0.70–1.5 mL/100 g (FS). For SS, the only salt in the mix formulation, the levels chosen to be added in wheat flour were those generally used by bread-making producers for sodium chloride when it is used in a bread recipe (up to 1.5%). Response surface methodology (RSM) is an empirical modeling statistical technique which has an important role in the design, development and proposal of new products to optimize the parameters and optimum condition of a process response variable [23,24]. Obtaining the optimal values using the response surface methodology involves three main steps: design of the experiment, obtaining the coefficients for the mathematical models and the prediction of the system responses by the design of experiment methodology (DOE), using the trial version of Design Expert software, (Stat-Ease, Minneapolis, MN, USA). In this research, three independent variables were used: *X*_1_—sea salt (A), *X*_2_—dry sourdough (B) and *X*_3_—fermented sugar (C). The effect of variation in their amounts on the rheological parameters of the dough (dependent variables—*Y*_1–13_) was studied. The three independent variables and their real and coded values used in the experimental design matrix, comprising 20 experiments, are shown in Table 1.

The rheological parameters determined with the Farinograph in this case were: WA—*Y*_1_; ST—*Y*_2_; DS—*Y*_3_. The rheological parameters determined with the Extensograph were: E—*Y*_4_; R_50_—*Y*_5_; Ext—*Y*_6_; R_max_—*Y*_7_.The rheological parameters determined with the Amylograph were: PV_max_—*Y*_8_; T_max_—*Y*_9_; H’m—*Y*_10_; VT—*Y*_11_; VR—*Y*_12_; CR—*Y*_13_. The rheological values obtained for the dough samples for the different levels of sea salt, dry sourdough and fermented sugar addition, according to our design, were performed twice. The average values were used in the statistical processing. 

The responses for the independent variables used in our experiment are shown in Table 2 and Table 3. For responses of dough rheological properties, experimental values were expressed as means ± standard deviations.

The responses of the system (*Y*_1__–13_) (Equation (1)) have been defined by the following mathematical model:
(1)$$Y=f\left(X_{1},X_{2},X_{3}\right)=\beta_{0}+\overset{3}{\underset{i=1}{\sum }}\beta_{i}\cdot X_{i}+\overset{3}{\underset{\begin{array}{c} i,j=1\\ i\neq j \end{array}}{\sum }}\beta_{ij}\cdot X_{i}\cdot X_{j}+\overset{3}{\underset{i=1}{\sum }}\beta_{ii}\cdot X_{i}^{2}$$
where: *β*_0_—constant coefficient; *β_i_*—linear coefficient; *β_ij_*—interaction coefficient; *β_ii_*—quadratic coefficient; *X_i_* and *X_j_*—the coded values of the independent variables. The ANOVA test was used to evaluate the significance of the mathematical model terms, as it compares the response variation with the variation due to random error at the 95% probability value (*p*-value). The suitability of the mathematical model has been checked by the Fisher tests (*F*-tests), and by the adjusted coefficient of determination (Adjusted *R*^2^), the accuracy of the fitted polynomial equation was determined. The non-significant coefficients were eliminated from the equations. In order to show the correlation between the independent and dependent variables, three-dimensional representations of the response surfaces were generated.

For responses and bakery tests, experimental values were expressed as means ± standard deviations. The determinations were performed twice. Statistical analysis was carried out using a trial version 12 of XLSTAT by an analysis of variance (one-way ANOVA) to evaluate the difference between means at *p* < 0.05 with Tukey’s test (at a 5% significance level).

## 3. Results

### 3.1. Fitting Models

Following the statistical processing of the experimental data, in order to show the effects of the independent variables on the predictive mathematical models for rheological properties during the mixing of sea salt (SS), dry sourdough (SD) and fermented sugar (FS) mixtures, the most fitting mathematical models (quadratic models) were obtained for the parameters of dough stability (ST), temperature at peak viscosity (T_max_), height under constraint of dough at maximum development time (H’m) and volume of the gas retained in the dough at the end of the test (VR).

### 3.2. Dough Rheological Properties during Mixing and Extension

The effect of SS, SD and FS addition in wheat flour on dough mixing properties on Farinograph values expressed by their quadratic models are shown in Table 4. The ANOVA results show that the obtained models were statistically significant (*p* < 0.001) for dough stability (ST), degree of softening at 10 min (DS) and water absorption (WA) with high Adjusted *R*^2^ values for ST and DS parameters. 

For the dough development time (DT) value, no significant model was obtained. A *p*-value of less than 0.05 indicates that model terms are significant. In our case, we obtained *p*-value = 0.84 (for the linear model), *p*-value = 0.93 for the Two Factor Interaction (2FI) model and *p*-value = 0.97 for the quadratic model. In this case, there are no significant model terms. Values of *p* greater than 0.1 indicate that the mathematical model terms are not significant. The most significant models were obtained for ST—Adjusted *R*^2^ = 0.78 and DS—Adjusted *R*^2^ = 0.82, followed by those obtained for WA for which Adjusted *R*^2^ = 0.62. The linear regression coefficients SS and FS presented a negative effect on WA, whereas the SD term presented a positive one on the WA value. The negative effect of SS on WA value has also been reported by different researchers, such as Lopes et al. [3], McCann and Day [25], Beck et al. [26], Beck et al. [27], Jekle et al. [28] and Uthayakumaran et al. [29], whereas the positive effect of SD has been previously reported by [14,21]. Chloride salts, just as in the case of SS, increased hydrophobic interactions between gluten proteins, which aggregate to a higher extent, decreasing the WA value. Furthermore, the FS presence may decrease the pH value, which will affect the positive charge of proteins, leading to changes in their conformation. This may favor a development of the unfold proteins, leading to a higher amount of reactive groups available to interact with water, a fact that may increase the WA value. On the ST value, the SS and FS have a positive effect, whereas the SD has a negative one, as it may be seen from Figure 1a,b. This indicates that SS and FS have a strengthening effect on wheat flour dough. A similar effect of chloride salts as SS on wheat flour dough has also been previously reported [29,30,31]. This is due to the surface hydrophobicity of the gluten proteins, which promotes a higher aggregation in the chloride salt presence [12], leading to higher ST values. Furthermore, in the presence of FS, which contains organic acids, some of the proteins are solubilized, thus leading to an increase in the osmotic pressure outside of the protein globules and causing dehydration of the gluten, which remains in an insoluble form, which makes it stronger. The negative effect of SD on ST may be due to its enzymatic activity, which may act on proteins, weakening the dough network. A similar behavior of SD on wheat flour dough has also been previously reported [14,21]. On DS value, the SS and FS presented a negative effect, whereas the SD showed a positive one, as it may be seen from Figure 1c,d. These independent variables’ effect on the DS response value confirms the behavior previously reported that SS and FS addition strengthens the dough, whereas SD weakens it.

The Extensograph values: energy (E), resistance to extension (R_50_), extensibility (Ext) and maximum resistance to extension (R_max_) and ratio number at a proving time of 135 min (R/E) were quadratically influenced by SS, SD and FS addition. All the quadratic models obtained for the Extensograph data were significant (*p* < 0.001) with an Adjusted *R*^2^ higher than 0.85 for all the dependent variables, as it may be seen from Table 2. The effect of SS and FS was a highly significant positive one (*p* < 0.01) on E, R_50_, R_max_ and R/E whereas SD had a negative effect on these variables. The effect of synergy action of independent variables SS-SD and SS-FS on the rheological parameters determined by the Extensograph led to obtaining of some quadratic models significant for the dependent variables E, R_max_, as it can be seen in Figure 2a–d.

The E, R_50_ and R_max_ measure the force required to stretch the dough, indicating the dough’s ability to resist deformation forces. Therefore, the higher these values are, the stronger the dough is. The ratio number R/E is also a measure of dough strength, meaning that higher values of this parameter indicate the fact that higher forces are required to stretch the dough. Therefore, the positive effect of SS and FS on these Extensograph values indicates their strengthening effect on wheat flour dough, whereas the negative effect of SD on these values indicates a weakening one. The strengthening effect produced by chloride salts, as SS on wheat flour dough was previously reported [12,25,32,33], whereas the weakening one of SD on wheat flour dough was reported by Nogueira et al. [14] and Voinea et al. [21]. The regression model for the Ext value has an *R*^2^ of 0.92, showing that the model could be used to explain more than 92% of the variability in the response. In this model, SS, SD and the interactions between SS and FS and SD and FS are significant model terms (*p* < 0.001). All the linear regression coefficients, SS, SD and FS added in wheat flour dough, presented a negative effect on this response, showing that dough extensibility decreased with the increased level of SS, SD and FS addition. Similar results were also obtained for SS addition in wheat flour [12] and for SD incorporation in dough recipe [14,21]. The negative effect of FS on Ext may be due to its organic acid content, knowing that normally, the acid addition in wheat flour decreases dough extensibility [34].

### 3.3. Dough Rheological Properties during Pasting

The results from model analysis on the Amylograph values showed a significant quadratic effect of SS, SD and FS on peak viscosity (PV_max_) and temperature at peak viscosity (T_max_), whereas on gelatinization temperature (T_g_, °C), no significant model was obtained. The mathematical models obtained for the gelatinization temperature (T_g_, °C) were not significant because we obtained the following values of *p*: *p*-value = 0.062 (for the linear model), *p*-value = 0.2243 (for the 2FI model) and *p*-value = 0.2282 (for the quadratic model). The experimental modelling analysis on Amylograph values is shown in Table 5. The most significant model for Amylograph was that obtained for T_max_, where Adjusted *R^2^* = 0.71, followed by those obtained for PV_max_ where Adjusted *R*^2^ = 0.65.

There was a positive effect on PV_max_ produced by the linear terms SS and FS, whereas a negative one was produced by the linear term SD and the interaction terms between the independent variables SS, SD and FS, as shown in Figure 3a,b.

Following SD and FS addition, the pH of the wheat flour dough decreased. This fact will influence the main components of the wheat flour dough, namely gluten and starch. SS is a product obtained through wheat fermentation. Furthermore, FS is a fermented sugar. It is well known that through the fermentation process, the amount of lactic bacteria and acidic products increases, which leads to a decrease in pH values. The PV_max_ decreased following the interaction between SS, SD and FS. This may be explained by the fact that, at low pH values specific to dough systems, by adding SD and FS, the solubility of the protein fractions are affected. Furthermore, a slightly acidic hydrolysis of wheat flour starch is known to take place under the pH conditions of acidic dough [35]. The structural changes from the acidic dough are also influenced by enzyme activity from the wheat flour, especially in amylases which affect starch rheological behavior during heating, as was recorded by the Amylograph device. However, the addition of SS in a linear form suppressed amylolytic activity due to the salt’s presence, which in turn led to an increase in the PV_max_ value. A similar behavior was previously reported by different researchers [14,21]. Furthermore, an increase in the PV_max_ value may also be noticed by adding FS in a linear form. This fact may be due to a decrease in the pH value of the dough system to a value that may reduce amylase activity in a significant way. However, the linear term SD and the interaction terms between SS, SD and FS decreased the PV_max_ value. This behavior may be attributed to the SD ingredient, which may contain a certain amount of α amylase activity, taking into account that SD is obtained after the fermentation process of wheat flour [21].

For the T_max_ value, a positive effect was produced by the linear terms related to SS, SD and the interactions between them, and a negative one by FS and the interaction between the latter and the independent variables SS and SD, as shown in Figure 3c,d. It is well known that the T_max_ value depends, among others, on the amount of water in the dough system. The SS, due to its ionic nature, decreased water activity, whereas SD addition favors protein weakening and, therefore, the water availability in the dough system. This influences the temperature value at which PV_max_ was recorded, these data being in accordance with those reported by Nogueira et al. [14] and Voinea et al. [21].

### 3.4. Dough Rheological Properties during Fermentation

The dough rheological parameters recorded during fermentation were maximum height of gaseous production (H’m), volume of the gas retained at the end of the test (VR), total CO_2_ volume production (VT) and retention coefficient (CR). In Table 3, the models obtained for these parameters and the regression coefficients are presented. For all the Rheofermentometer values, the most significant influences were presented by the linear term FS and the interaction between SS and FS (*p* < 0.01). The H’m, VR and CR were quadratic, being influenced by the amount of SS, SD and FS added. All the models obtained for the Rheofermentometer data were significant (*p* < 0.01), which reflects the fact that they are useful to describe the relationship between the SS, SD, FS and the dependent variables. According to the Adjusted *R*^2^ values, the most significant mathematical models were those obtained for VT (Adjusted *R*^2^ = 0.80) and CR (Adjusted *R*^2^ = 0.83), followed by those obtained for VR (Adjusted *R*^2^ = 0.77) and H’m (Adjusted *R*^2^ = 0.75). The effects of the interaction factors (SS–SD; SS–FS) obtained for the Rheofermentometer values VT and CR are shown in Figure 4.

The independent variable addition of SS and SD presented a positive effect on H’m, VT and VR values, whereas addition of FS presented a negative one. The positive effect of SS and SD parameters on these Rheofermentometer values may be explained by an increase in the yeast fermentation speed due to SS and SD addition [34]. This caused an increase in the gas formed during fermentation and, therefore, in the VT value, which consequently led to an increased H’m value. Generally speaking, it is well known that chloride salts affect yeast fermentation, especially by repressing it, due to the osmotic pressure effect [35]. However, when low levels of salt are incorporated in the dough system, yeast cell multiplication is stimulated. This behavior is attributed to the annihilation of the toxic action of thionine by salt, which slows down yeast fermentation activity [21]. According to our design from the 20 runs analyzed, only five of them exceeded the level of 1 g/100 g salt addition in wheat flour. On the other hand, it is well known that only levels higher than 1% have a significant effect on suppressing yeast fermentation activity. Therefore, taking into account that 75% of the samples analyzed in the present study presented low levels of salt incorporated in the dough system, the general effect was a positive one on H’m, VT and VR values, due to increased yeast fermentation activity. Regarding the positive effect of SD on H’m, VT and VR values, this may be due to its amylolytic activity, which favors starch hydrolysis, leading to more fermentable sugars available for yeast activity [36,37]. From the independent variables used, only FS presented a negative effect on H’m, VT and VR values. This has also presented the most significant influence (*p* < 0.01) on these Rheofermentometer values. This may be due to the fact that this ingredient is a fermented sugar product which contains organic acids that may lead to a decrease in the pH dough value. This may cause a decrease in α-amylase activity, which may lead to lower levels of fermentable sugars for yeast activity. As a consequence, the gas formed during fermentation will decrease, resulting in lower H’m, VT and VR values. For CR value, SS and SD presented a negative effect, whereas FS presented a positive one. Taking into account that CR is the ratio between VR and VT values, it is possible for FS to have a higher effect on VR value than SS and SD on the VT one. 

### 3.5. Optimization of Sea Salt, Dry Sourdough and Fermented Sugar Formulation

Obtaining the optimal values of the dough rheological parameters was one of the objectives of this study. Based on the results obtained in the experiment, the combination of the response surface methodology with Derringer’s desirability function (Equation (2)) allows the prediction of the optimal values of independent variables. The Derringer’s desirability function approach is one of the most used tools in industry for optimizing multiple response processes [20,38]. Derringer’s desirability function is based on the idea that the quality of a process that has several quality characteristics, with some of them outside desired limits, is completely undesirable.
(2)$$D={\left(d_{1}^{r_{1}}\cdot d_{2}^{r_{2}}\cdot \ldots \cdot d_{n}^{r_{n}}\right)}^{\frac{1}{\sum r_{i}}}$$
where *d*_1_, *d*_2_, …, *d_n_* are indices of desirability of the dependent variables and *r*_1_, *r*_2_, …, *r_i_* are the relative importance of the dependent variables. The desirability function takes values between 0 and 1, 0 for undesirable values, and 1 for a completely desirable value. The optimization process was performed using the trial version 12 of Design Expert software, (Stat-Ease, Minneapolis, MN, USA). Thus, applying the desirability function methodology, the optimal values of the independent variables were calculated (Figure 5).

The optimum amounts obtained were 0.30 g/100 g SS, 0.50 g/100 g SD and 1.02 mL/100 g FS added in wheat flour, with a desirable function score of 0.618018. Compared to our previous results [20,21] when other formulations for sodium replacement were used, we may conclude that this was the best one. This formulation contains the lowest sodium amount from sea salt composition (only 0.30% compared to 1.396% SS in our previous study [21] or 1.31% NaCl of another study [20]. The corresponding responses for these optimum values obtained in this study are shown in Table 6, in which these data are compared to those in which 1.5% NaCl was added in the wheat flour (control sample). 

It may be seen that dough rheological properties during mixing and pasting are not significantly different between the optimized and control sample. Regarding the extension data obtained through the Extensograph device, similar data may be noticed, with a significant difference only between maximum resistance and ratio number at a proving time of 135 min, for which the optimized sample presented higher values. This indicates that the optimum formulation of SS, SD and FS has more of a strengthening effect on dough rheological properties during extension than the chloride salt at a 1.5% addition in wheat flour. Regarding dough rheological properties during fermentation, a slight difference may be seen between the compared samples, the optimized one presenting higher values. This behavior may be due to the fact that a higher level of NaCl addition (such as 1.5% NaCl addition in wheat flour) causes yeast activity to decrease on the semi-permeable membrane of yeast cells and to the osmotic pressure from the dough system [21]. Furthermore, the optimized sample contains dry sourdough (SD), which may exhibit slightly amylolytic activity, favoring starch hydrolysis, which increases yeast activity [37], and sea salt in a low amount of only 0.30 g/100 g wheat flour, which, at this level, stimulates yeast activity [21]. 

### 3.6. Quality Characteristics for Bread Samples with 1.5% NaCl Addition (Control Sample) and the Optimized Sea Salt, Sry Sourdough and Sermented Sugar Formulation

The bread quality characteristics for the optimized and control bread samples are shown in Table 7. Furthermore, the image of the obtained bread samples is shown in Figure 6.

As it may be seen, no significant differences (*p* < 0.05) were obtained between samples. However, it may be noticed that the optimized sample presented a higher quality than the control one. It presented higher physical and textural characteristics (with the exception of firmness) and was better appreciated from the sensory point of view. The higher physical and textural characteristics of the optimized bread sample may be due to the fact that it was obtained from wheat flour dough with higher values for dough rheological properties during fermentation, such as maximum height of gaseous production, total CO_2_ volume production, volume of the gas retained in the dough at the end of the test than the control sample. Therefore, the optimized bread sample (Figure 6a) was obtained from a wheat flour dough which can better retain the gas formed in a higher amount during the fermentation process than the control sample (Figure 6b).

From the color point of view, it may be seen that the optimized sample presents lower lightness (*L** value) and higher greenness (*a** value) and yellowness (*b** value) than the control sample. This may be attributed to the formulation mix used to obtain the optimized sample. It contains dry sourdough, which may present a slightly amylolytic and proteolytic activity. This increases the amount of amino acids and fermentable sugars from the dough system, favoring melanoidin formation during the baking process, leading to a darker bread color. This may also contribute to the formation of a higher amount of flavor compounds during baking, which improves the bread’s sensory properties. Apart from these properties, the optimized sample contains fermented sugar, which presents aroma components that lead to a slightly acidic flavor combined with ferment and savory notes (bouillon, umami). The data obtained for the optimized bread sample from the sensory point of view were quite surprising for us since this sample contains a very low amount of sodium chloride, which comes only from sea salt. Even so, the taste and flavor of the optimized sample were better appreciated than those of the control one. 

## 4. Conclusions

According to the data obtained, all of the independent variables, SS, SD and FS, changed the rheological characteristics of wheat flour dough during mixing, extension, pasting and fermentation. During mixing, it was noticed that water absorption (WA) decreased by SS and FS addition and increased when SD was incorporated in the wheat flour dough. From all the independent variables used, it seems that only SD presented a significant effect on dough weakening by decreasing dough stability (ST) (*p* < 0.1) and increasing the degree of softening at 10 min DS (*p* < 0.01). As well as that, SD also presented a significant (*p* < 0.1) weakening effect on dough rheological properties during extension by decreasing all the Extensograph E, R_50_ and R_max_ values, whereas the SS and FS increase was significant (*p* < 0.01). On peak viscosity during pasting, SS and FS presented a significantly positive effect (*p* < 0.01), whereas SD presented a negative one. On all the Rheofermentometer-analyzed values except the retention coefficient value, the SS and SD presented a positive effect, whereas FS, a negative one. Response surface methodology was an efficient statistical tool able to model the influence of SS, SD, and FS on dough rheological properties. The models obtained for the variables were significant, with high values of Adjusted *R*^2^ ≥ 0.71 (except for WA—0.62 and PV_max_—0.65) and all *p*-values < 0.01 showing, for most dependent variables, no lack of fit. The optimum values, obtained with the numerical method, were: for SS—0.30 g/100 g wheat flour, for SD—0.50 g/100 g wheat flour and for FS—1.02 mL/100 g wheat flour. For this formulation, compared to a control sample in which only 1.5% NaCl was incorporated in wheat flour, the dough rheological properties during fermentation were higher but no significant differences were obtained for those obtained during mixing and pasting. The comparison between the optimized bread sample characteristics obtained through the baking test and control sample indicates no significant differences (*p* < 0.05) between samples. However, the optimized sample presented better characteristics from the physical and textural points of view and was darker and more appreciated by the panelists from the sensory point of view.

## Figures and Tables

**Figure 1 foods-09-01465-f001:**
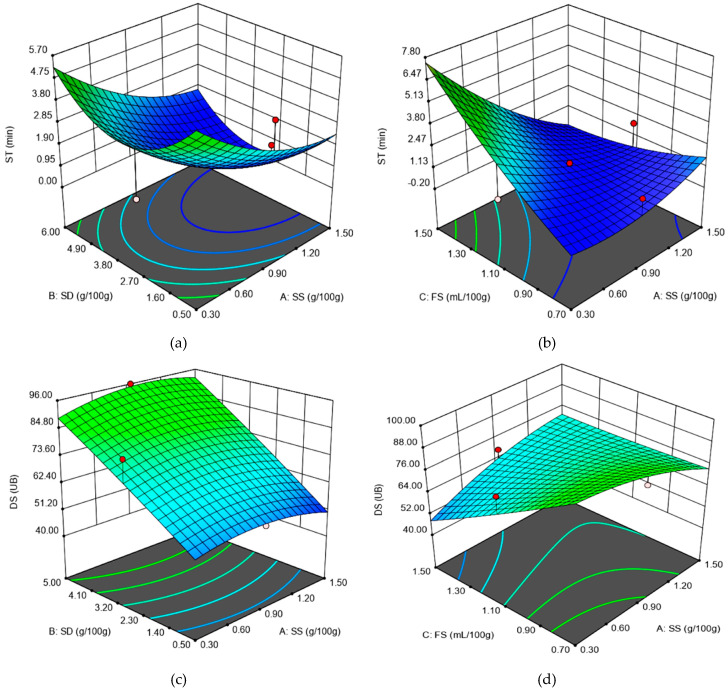
The graphical representations of the Farinograph parameters: (**a**) stability (ST) as affected by the levels of sea salt (SS) and dry sourdough (SD) incorporated in wheat flour at 1.10 (mL/100 g) fermented sugar; (**b**) stability (ST) as affected by the levels of sea salt (SS) and fermented sugar (FS) incorporated in wheat flour at 2.75 (g/100 g) dry sourdough; (**c**) degree of softening at 10 min (DS) as affected by the levels of sea salt (SS) and dry sourdough (SD) incorporated in wheat flour at 1.10 (mL/100 g) fermented sugar; (**d**) degree of softening at 10 min (DS) as affected by the levels of sea salt (SS) and fermented sugar (FS) incorporated in wheat flour at 2.75 (g/100 g) dry sourdough.

**Figure 2 foods-09-01465-f002:**
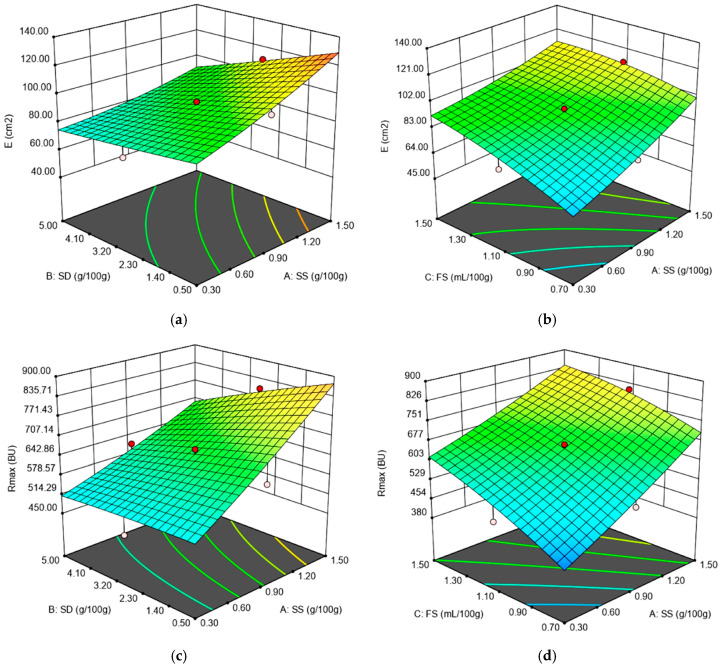
Graphical representations of the Extensograph parameters: (**a**) energy (E) as affected by the levels of sea salt (SS) and dry sourdough (SD) incorporated in wheat flour at 1.10 (mL/100 g) fermented sugar; (**b**) energy (E) as affected by the levels of sea salt (SS) and fermented sugar (FS) incorporated in wheat flour at 2.75 (g/100 g) dry sourdough; (**c**) maximum resistance (R_max_) as affected by the levels of sea salt (SS) and dry sourdough (SD) incorporated in wheat flour at 1.10 (mL/100 g) fermented sugar; (**d**) maximum resistance (R_max_) as affected by the levels of sea salt (SS) and fermented sugar (FS) incorporated in wheat flour at 2.75 (g/100 g) dry sourdough.

**Figure 3 foods-09-01465-f003:**
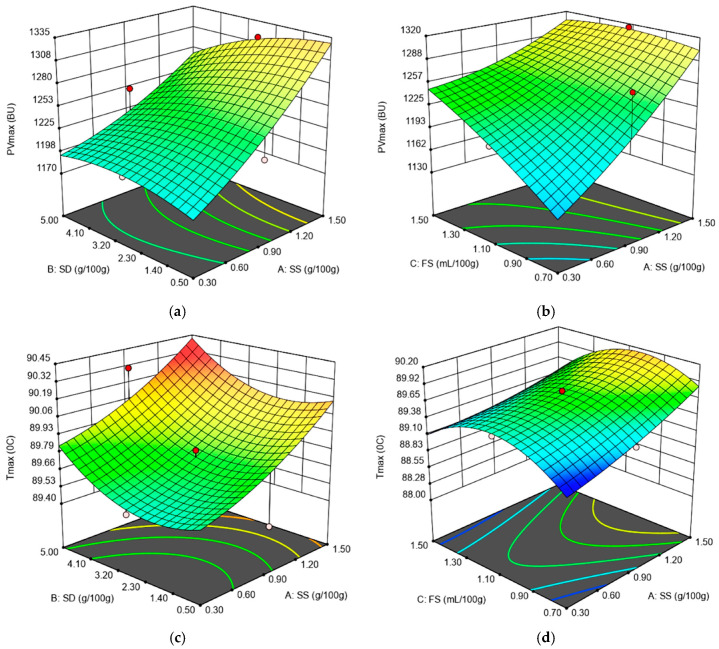
Graphical representations of the Amylograph parameters: (**a**) peak viscosity (PV_max_) as affected by the levels of sea salt (SS) and dry sourdough (SD) incorporated in wheat flour at 1.10 (mL/100 g) fermented sugar; (**b**) peak viscosity (PV_max_) as affected by the levels of sea salt (SS) and fermented sugar (FS) incorporated in wheat flour at 2.75 (g/100 g) dry sourdough; (**c**) temperature at peak viscosity (T_max_) as affected by the levels of sea salt (SS) and dry sourdough (SD) incorporated in wheat flour at 1.10 (mL/100 g) fermented sugar; (**d**) temperature at peak viscosity (T_max_) as affected by the levels of sea salt (SS) and fermented sugar (FS) incorporated in wheat flour at 2.75 (g/100 g) dry sourdough.

**Figure 4 foods-09-01465-f004:**
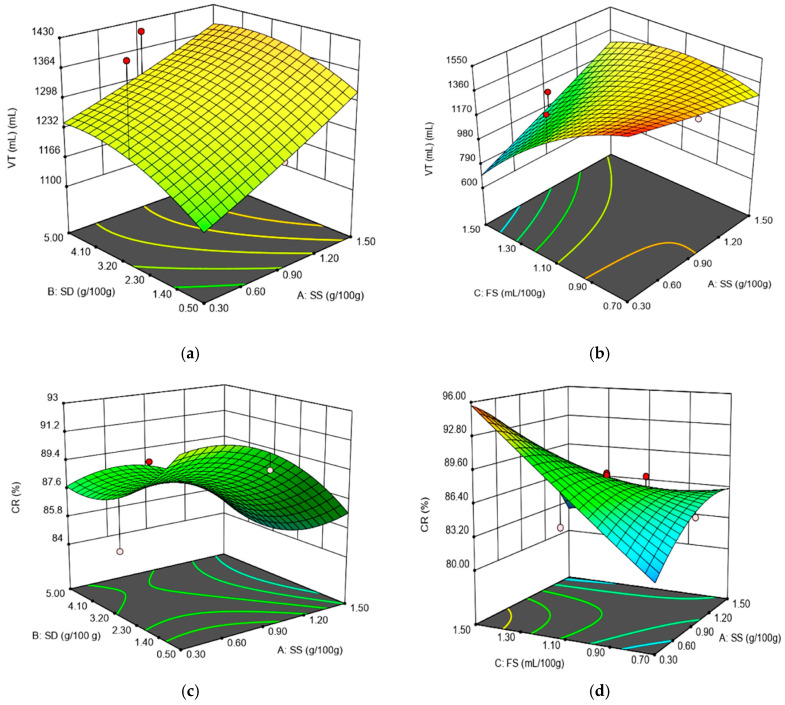
Graphical representations of the Rheofermentometer parameters: (**a**) total CO_2_ volume production (VT) as affected by the levels of sea salt (SS) and dry sourdough (SD) incorporated in wheat flour at 1.10 (mL/100 g) fermented sugar; (**b**) total CO_2_ volume production (VT) as affected by the levels of sea salt (SS) and fermented sugar (FS) incorporated in wheat flour at 2.75 (g/100 g) dry sourdough; (**c**) retention coefficient (CR) as affected by the levels of sea salt (SS) and dry sourdough (SD) incorporated in wheat flour at 1.10 (mL/100 g) fermented sugar; (**d**) retention coefficient (CR) as affected by the levels of sea salt (SS) and fermented sugar (FS) incorporated in wheat flour at 2.75 (g/100 g) dry sourdough.

**Figure 5 foods-09-01465-f005:**
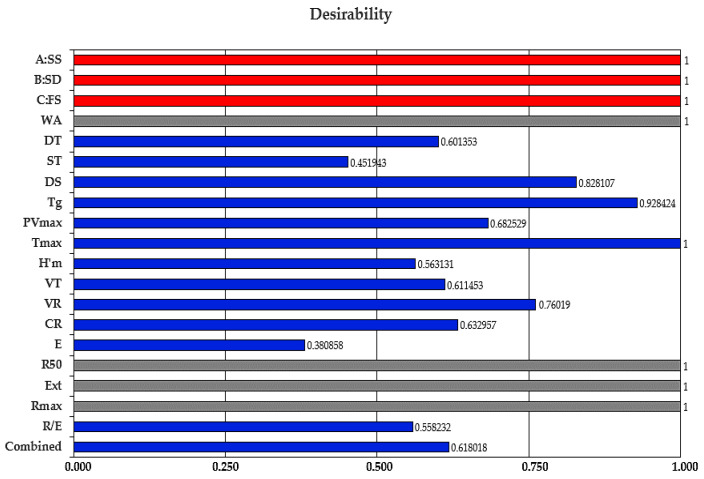
Desirability function scores for the studied independent and dependent variables: sea salt (SS), dry sourdough (SD), fermented sugar (FS), water absorption (WA), dough development time (DT), dough stability (ST), degree of softening at 10 min (DS), gelatinization temperature (T_g_), peak viscosity (PV_max_), temperature at peak viscosity (T_max_), maximum height of gaseous production (H’m), total CO_2_ volume production (VT), volume of the gas retained in the dough at the end of the test (VR), retention coefficient (CR), energy (E), resistance to extension (R_50_), extensibility (Ext), maximum resistance to extension (R_max_) and ratio number (R/E) at a proving time of 135 min.

**Figure 6 foods-09-01465-f006:**
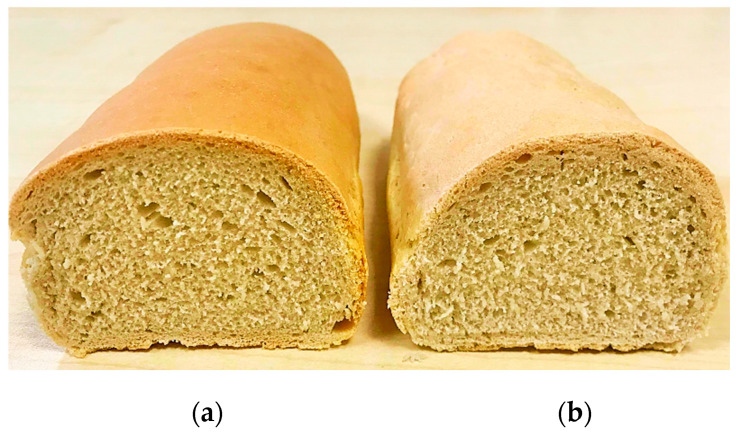
The bread samples obtained: (**a**) optimized bread sample, (**b**) control bread sample.

**Table 1 foods-09-01465-t001:** Experimental design with real values and coded values of independent variables.

Run	Real Values			Coded Values		
	SS (g/100 g)	SD (g/100 g)	FS (mL/100 g)	*X* _1_	*X* _2_	*X* _3_
1	0.90	2.75	1.10	0	0	0
2	1.50	0.50	1.50	+1	−1	+1
3	0.90	2.75	1.10	0	0	0
4	0.30	5.00	1.50	−1	+1	+1
5	0.90	0.50	1.10	0	−1	0
6	0.90	2.75	1.50	0	0	+1
7	1.50	5.00	0.70	+1	+1	−1
8	1.50	5.00	1.50	+1	+1	+1
9	0.30	0.50	0.70	−1	−1	−1
10	0.30	2.75	1.10	−1	0	0
11	0.30	0.50	1.50	−1	−1	+1
12	0.30	5.00	0.70	−1	+1	−1
13	1.50	0.50	0.70	+1	−1	−1
14	0.90	2.75	0.70	0	0	−1
15	0.90	2.75	1.10	0	0	0
16	0.90	2.75	1.10	0	0	0
17	0.90	5.00	1.10	0	+1	0
18	0.90	2.75	1.10	0	0	0
19	0.90	2.75	1.10	0	0	0
20	1.50	2.75	1.10	+1	0	0

**Table 2 foods-09-01465-t002:** The effects of different levels of independent variable addition on dough rheological properties during mixing and extension of sea salt–dry sourdough–fermented sugar mixtures.

Run	Farinograph			Extensograph (Proving Time 135 min)				
	WA (%)	ST (min)	DS (UB)	E (cm^2^)	R_50_ (BU)	Ext (mm)	R_max_ (BU)	R/E
1	58.0 ± 0.41	1.4 ± 0.10	76 ± 1.12	95 ± 2.3	611 ± 5.1	110 ± 1.3	665 ± 5.8	6.0 ± 0.14
2	55.7 ± 0.70	1.2 ± 0.10	51 ± 0.80	132 ± 4.1	898 ± 8.9	106 ± 1.2	949 ± 9.7	6.8 ± 0.16
3	58.0 ± 0.41	1.4 ± 0.10	76 ± 1.12	95 ± 2.3	611 ± 5.1	110 ± 1.3	665 ± 5.8	6.0 ± 0.14
4	57.4 ± 0.37	9.6 ± 0.14	44 ± 0.90	95 ± 2.3	545 ± 4.8	116 ± 1.4	627 ± 5.4	5.7 ± 0.13
5	56.4 ± 0.28	3.4 ± 0.14	55 ± 0.80	102 ± 3.0	530 ± 4.7	122 ± 1.1	629 ± 5.4	5.2 ± 0.12
6	59.2 ± 0.28	1.3 ± 0.14	69 ± 1.00	95 ± 2.3	619 ± 5.2	109 ± 1.3	693 ± 6.0	6.4 ± 0.17
7	59.8 ± 0.32	1.0 ± 0.14	97 ± 1.60	82 ± 1.7	590 ± 5.0	103 ± 1.3	620 ± 5.9	6.1 ± 0.13
8	59.6 ± 0.14	0.9 ± 0.14	93 ± 1.60	93 ± 2.3	680 ± 6.2	102 ± 1.3	711 ± 7.8	7.0 ± 0.24
9	57.8 ± 0.13	2.9 ± 0.14	59 ± 0.80	77 ± 2.2	392 ± 5.6	123 ± 1.5	453 ± 4.2	3.7 ± 0.10
10	60.0 ± 0.21	1.4 ± 0.10	81 ± 1.40	73 ± 2.0	430 ± 4.1	117 ± 1.4	466 ± 4.4	4.0 ± 0.10
11	57.4 ± 0.14	9.6 ± 0.14	44 ± 0.80	95 ± 2.3	545 ± 4.8	116 ± 1.4	627 ± 5.4	5.7 ± 0.13
12	61.4 ± 0.14	0.9 ± 0.14	130 ± 2.20	51 ± 0.8	354 ± 5.2	104 ± 1.4	364 ± 5.3	3.5 ± 0.10
13	57.4 ± 0.28	3.5 ± 0.14	56 ± 0.80	119 ± 3.6	671 ± 5.9	119 ± 1.5	784 ± 8.3	6.6 ± 0.18
14	58.9 ± 0.42	1.4 ± 0.10	85 ± 1.40	77 ± 2.2	462 ± 4.4	113 ± 1.3	520 ± 4.6	4.6 ± 0.10
15	58.0 ± 0.28	1.4 ± 0.10	76 ± 1.12	95 ± 2.3	611 ± 5.1	110 ± 1.3	665 ± 5.8	6.0 ± 0.14
16	58.0 ± 0.28	1.4 ± 0.10	76 ± 1.12	95 ± 2.3	611 ± 5.1	110 ± 1.3	665 ± 5.8	6.0 ± 0.14
17	60.2 ± 0.14	0.9 ± 0.03	96 ± 1.60	78 ± 2.2	573 ± 4.8	102 ± 1.3	612 ± 5.1	6.0 ± 0.14
18	58.0 ± 0.14	1.4 ± 0.10	76 ± 1.12	95 ± 2.3	611 ± 5.1	110 ± 1.3	665 ± 5.8	6.0 ± 0.14
19	58.0 ± 0.28	1.4 ± 0.10	76 ± 1.12	95 ± 2.3	611 ± 5.1	110 ± 1.3	665 ± 5.8	6.0 ± 0.14
20	57.8 ± 0.28	1.9 ± 0.14	63 ± 0.90	111 ± 3.4	730 ± 8.1	107 ± 1.2	803 ± 8.5	7.5 ± 0.25

WA—water absorption; ST—dough stability; DS—degree of softening at 10 min; E—energy; R_50_—resistance to extension up to 50 mm; Ext—extensibility; R_max_—maximum resistance; R/E—ratio number at a proving time of 135 min.

**Table 3 foods-09-01465-t003:** The effects of different levels of independent variable addition on dough rheological properties during pasting and fermentation of sea salt–dry sourdough–fermented sugar mixtures.

Run	Responses					
	PV_max_ (BU)	T_max_ (℃)	H’m (mm)	VT (mL)	VR (mL)	CR (%)
1	1245 ± 5.66	89.8 ± 0.3	72.5 ± 1.1	1267 ± 6.7	1122 ± 8.1	88.5 ± 0.9
2	1368 ± 1.41	89.8 ± 0.3	55.1 ± 0.7	1213 ± 6.5	1023 ± 7.8	84.3 ± 0.7
3	1245 ± 2.83	89.8 ± 0.3	72.4 ± 1.1	1266 ± 6.6	1120 ± 8.0	88.4 ± 0.9
4	1251 ± 1.84	89.3 ± 0.2	37.2 ± 0.8	601 ± 4.2	589 ± 5.2	98.0 ± 1.2
5	1218 ± 1.41	89.4 ± 0.3	62.5 ± 1.1	1222 ± 8.3	1104 ± 7.8	90.3 ± 1.0
6	1207 ± 1.13	89.1 ± 0.3	70.6 ± 1.0	1103 ± 9.6	1002 ± 7.2	90.8 ± 0.7
7	1265 ± 2.12	90.3 ± 0.4	64.0 ± 0.85	1301 ± 4.0	1119 ± 8.3	86.0 ± 0.5
8	1252 ± 2.10	89.5 ± 0.35	60.0 ± 0.8	1272 ± 6.7	1049 ± 8.1	82.4 ± 0.3
9	1113 ± 1.10	89.1 ± 0.3	70.6 ± 0.9	1376 ± 6.8	1213 ± 6.6	88.1 ± 0.4
10	1198 ± 1.15	89.5 ± 0.4	75.6 ± 1.0	1427 ± 6.9	1215 ± 6.8	85.1 ± 0.4
11	1251 ± 2.10	89.3 ± 0.4	37.2 ± 0.5	601 ± 3.1	589 ± 3.4	98.0 ± 0.2
12	1107 ± 1.20	89.3 ± 0.4	81.0 ± 1.5	1538 ± 8.1	1216 ± 6.5	79.0 ± 0.3
13	1286 ± 2.30	89.8 ± 0.4	64.1 ± 0.8	1279 ± 6.8	1132 ± 7.9	88.5 ± 0.6
14	1270 ± 2.13	89.3 ± 0.4	72.3 ± 1.1	1364 ± 6.7	1161 ± 6.3	85.1 ± 0.6
15	1245 ± 2.10	89.8 ± 0.5	72.5 ± 1.1	1267 ± 6.7	1122 ± 7.3	88.5 ± 0.7
16	1245 ± 2.10	89.8 ± 0.5	72.3 ± 1.1	1264 ± 6.2	1120 ± 7.2	88.6 ± 0.7
17	1250 ± 2.13	90.3 ± 0.6	75.9 ± 1.2	1405 ± 5.9	1244 ± 6.8	88.5 ± 0.7
18	1245 ± 2.10	89.8 ± 0.5	72.4 ± 1.1	1266 ± 8.4	1120 ± 7.2	88.4 ± 0.5
19	1245 ± 2.10	89.8 ± 0.5	72.3 ± 1.1	1267 ± 7.9	1121 ± 7.2	88.4 ± 0.7
20	1315 ± 2.83	90.0 ± 0.5	67.2 ± 1.0	1276 ± 8.1	1108 ± 6.8	86.8 ± 0.7

PV_max_—peak viscosity, T_max_—temperature at peak viscosity, H’m—height under constraint of dough at maximum development time, VT—total volume of CO_2_ produced during fermentation, VR—volume of the gas retained in the dough at the end of the test, CR—retention coefficient.

**Table 4 foods-09-01465-t004:** The effects of independent variables (SS, SD and FS), expressed as their coefficients on the predictive mathematical models for the rheological parameters of the dough during mixing and extension of sea salt–dry sourdough–fermented sugar mixtures.

Factors ^b^	Parameters							
	Farinograph			Extensograph (Proving Time 135 min)				
	WA (%)	ST (min)	DS (UB)	E (cm^2^)	R_50_ (BU)	Ext (mm)	R_max_ (BU)	R/E
Constant	+58.28	+1.20	+76.11	+92.50	+591.45	+110.95	+648.95	+5.89
*X* _1_	−0.37	+1.59 **	+0.20	+14.60 **	+130.30 **	−3.90 **	+133.00 **	+1.14
*X* _2_	+1.37 **	−0.73	+19.50 **	−12.60 **	−29.40 *	−5.90 **	−50.80 **	+0.03
*X* _3_	−0.60 *	+1.29	−12.60 **	+10.40 **	+81.80 **	−1.30 *	+86.60 **	+0.71
*X*_1_ × *X*_2_	+0.34	−0.10	+1.50	−6.25 **	−32.62	−0.12	−39.12 **	−0.012
*X*_1_ × *X*_3_	+0.31	−2.22 **	+11.50 **	−4.75 *	−3.37	−2.37 **	−22.62	−0.39
*X*_2_ × *X*_3_	−0.26	+0.53	−8.75 *	+3.00	−12.37	+3.88 **	+1.88	+0.11
*X* _1_ ^2^	+0.20	+0.74	−4.27	+2.45	+17.86	+0.59	+9.64	+0.036
*X* _2_ ^2^	−0.40	+1.24	−0.77	+0.45	−10.64	+0.59	−4.36	−0.11
*X* _3_ ^2^	+0.35	+0.44	+0.73	−3.55	−21.64	−0.41	−18.36	−0.21
Adjusted *R*^2^	0.62	0.78	0.82	0.92	0.87	0.85	0.90	0.85
Std. dev.	0.91	1.20	8.81	5.00	43.80	2.43	40.33	0.41
*p*-value	0.0003 **	0.001 **	<0.0001 **	<0.0001 **	<0.0001 **	0.0002 **	<0.0001 **	0.0002 **

^a^ Significant at *p* < 0.01 **, at *p* < 0.05 *. ^b^
*X*_1_—SS (g/100 g); *X*_2_—SD (g/100 g); *X*_3_—FS (mL/100 g); Adjusted *R*^2^ is measure of fit of the model; Std. dev. is standard deviation. WA—water absorption; ST—dough stability; DS—degree of softening at 10 min; E—energy; R_50_—resistance to extension up to 50 mm; Ext—extensibility; Rmax—maximum resistance; R/E—ratio number at a proving time of 135 min.

**Table 5 foods-09-01465-t005:** The effects of independent variables (SS, SD and FS), expressed as their coefficients on the predictive mathematical models for the rheological parameters of the dough during pasting and fermentation of sea salt–dry sourdough–fermented sugar mixtures.

Factors ^b^	Parameters					
	PV_max_ (BU)	T_max_ (°C)	H’m (mm)	VT (mL)	VR (mL)	CR (%)
Constant	+1241.35	+89.75	+73.65	+1229.95	+1147.61	+88.22
*X* _1_	+56.60 **	+0.29 **	+0.88	+79.80 *	+60.90 *	−2.02 **
*X* _2_	−11.10	+0.13 *	+2.86	+42.60	+15.60	−1.53 *
*X* _3_	+28.80 *	−0.08	−9.19 **	−206.80 **	−158.90 **	+2.68 **
*X*_1_ × *X*_2_	−16.37	+0.001	−0.70	−10.12	+1.25	+0.59
*X*_1_ × *X*_3_	−26.62	−0.12 *	+8.03 **	+202.13 **	+134.00 **	−4.59 **
*X*_2_ × *X*_3_	−11.12	−0.12 *	−0.67	−15.62	+4.50	+1.21 *
*X* _1_ ^2^	+10.64	+0.077	−4.13	+5.86	−26.27	−1.91
*X* _2_ ^2^	−11.86	+0.18	−6.33 *	−32.14	−13.77	+1.54
*X* _3_ ^2^	−7.36	−0.47 *	−4.08	−112.14	−106.27	+0.091
Adjusted *R*^2^	0.65	0.71	0.75	0.80	0.77	0.83
Std. dev.	34.95	0.19	5.81	103.26	84.20	1.79
*p*-value	0.00123 **	0.0045 **	0.0022 **	0.0007 **	0.0014 **	0.0003 **

^a^ Significant at *p* < 0.01 **, at *p* < 0.05 *. ^b^
*X*_1_—SS (g/100 g); *X*_2_—SD (g/100 g); *X*_3_—FS (mL/100 g); Adjusted *R*^2^ is measure of fit of the model; Std. dev. is standard deviation. PV_max_—peak viscosity, T_max_—temperature at peak viscosity, H’m—height under constraint of dough at maximum development time, VT—total volume of CO_2_ produced during fermentation, VR—volume of the gas retained in the dough at the end of the test, CR—retention coefficient.

**Table 6 foods-09-01465-t006:** The characterization of wheat flour dough rheological properties with 1.5% NaCl addition (control sample) and the optimized sea salt, dry sourdough and fermented sugar formulation sample.

Parameters	Values	
	Control Sample	Optimized Sample
Water absorption (%)	57.2 ± 0.1	56.6
Dough development time (min)	2.7 ± 0.1	1.5
Stability (min)	3.6 ± 0.2	4.6
Degree of softening (UB)	61 ± 0.8	57
Energy (cm^2^)	85 ± 2.2	83.95
Resistance to extension up to 50 mm (BU)	416 ± 3.9	437.35
Extensibility (mm)	122 ± 1.1	122.14
Maximum resistance (BU)	298 ± 5.6	502.69
Ratio number at a proving time of 135	2.78 ± 0.2	4.34
Gelatinization temperature (˚C)	63.9 ± 0.1	63.86
Peak viscosity (BU)	1290 ± 3.1	1163.81
Temperature at peak viscosity (˚C)	89.9 ± 0.2	89.50
Maximum height of gaseous production (mm)	50.6 ± 0.8	62.91
Total CO_2_ volume production (mL)	939 ± 3.5	1236.85
Volume of the gas retained in the dough at the end of the test (mL)	855 ± 2.3	1104.63
Retention coefficient (%)	91.0 ± 0.1	90.37

**Table 7 foods-09-01465-t007:** The characterization of bread quality characteristics for the samples with 1.5% NaCl addition (control sample) and the optimized sea salt, dry sourdough and fermented sugar formulation sample.

Parameters	Values	
	Control Sample	Optimized Sample
Physical characteristics		
Loaf volume (cm^3^/100 g)	332.97 ± 1.2 ^a^	343.83 ± 1.3 ^a^
Porosity (%)	95.00 ± 0.15 ^a^	96.66 ± 0.14 ^a^
Elasticity (%)	67.88 ± 0.12 ^a^	75.49 ± 0.15 ^a^
Color characteristics		
Lightness—*L**	73.9 ± 0.12 ^a^	71.81 ± 0.16 ^a^
Redness/greenness—*a**	−3.73 ± 0.08 ^a^	−3.72 ± 0.06 ^a^
Yellowness/blueness—*b**	20.83 ± 0.09 ^a^	21.24 ± 0.11 ^a^
Textural characteristics		
Springiness (%)	81.36 ± 0.22 ^a^	88.83 ± 0.12 ^a^
Cohesiveness	0.75 ± 0.01 ^a^	0.82 ± 0.02 ^a^
Gumminess (kg)	1.22 ± 0.06 ^a^	1.27 ± 0.05 ^a^
Firmness (kg)	1.61 ± 0.02 ^a^	1.55 ± 0.02 ^a^
Resilience	1.53 ± 0.06 ^a^	1.71 ± 0.04 ^a^
Sensory characteristics		
Appearance	8.44 ± 0.05 ^a^	8.66 ± 0.03 ^a^
Color	8.11 ± 0.07 ^a^	8.55 ± 0.06 ^a^
Taste	7.77 ± 0.12 ^a^	8.33 ± 0.17 ^a^
Smell	7.66 ± 0.16 ^a^	8.22 ± 0.11 ^a^
Texture	7.00 ± 0.21 ^a^	8.33 ± 0.18 ^a^
Flavor	7.44 ± 0.13 ^a^	8.11 ± 0.14 ^a^
Overall acceptability	8.00 ± 0.14 ^a^	8.44 ± 0.12 ^a^

The values are means ± standard deviations of three replicates. Means in the same column followed by identical superscript letters (a) indicates no significant difference at *p* < 0.05.

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
