# Peer review of "The Effect of Sea Salt, Dry Sourdough and Fermented Sugar as Sodium Chloride Replacers on Rheological Behavior of Wheat Flour Dough"

_foods, 2020, doi:10.3390/foods9101465_

Round 1

Reviewer 1 Report

This paper represents much hard work to utilise three different ingredients used at differing levels so that statistical designs can be used to give a model of use of materials. The driver for the work is very relevant and is based on the requirement for sodium reduction in baked products. The work seems to be carried out to a high standard and is reasonably described. There is some odd use of English, but the paper is understandable and generally well presented.

Despite the quantity and quality of the experimental work done, I do have major concerns on the publishing of this paper as I do not see the point of all this effort. The ingredients used are not well defined and therefore would be hard to reproduce. For example, it is stated that a low sodium content sea salt is used, this material has high levels of potassium and magnesium. However, fifty percent of the content within the salt is not defined. There is insufficient description of the sourdough and fermented sugar ingredients.  The paper deals with the rheological properties of the dough and how it leavens, however the statements at the start of the paper makes it clear that the materials being studied are added as “improvers of bakery flavour perception”.

The reasoning for the complex combined use of these poorly defined flavour enhancing materials to predict rheological performance is not well explained. Surely their actual use will be dependent on the flavour they can impart, and although I understand that the rheology of the samples is important in its own right and impacts on the perceived taste of the product, a better explanation of the relevance of the chosen additives and their levels is required. There is some excellent quoting of the literature that gives details of the relationship between ingredients and the resultant functionality, but it feels that known ideas are being retrospectively fitted to the results rather than creating any new understanding. I would not have thought new insights were possible without knowing much more about the ingredients.

The authors explain the statistical model and the fit of the data, but there is no indication on whether the values are appropriate or not. There is no reason to think that many of these rheological parameters are linearly related to salt levels etc. I don’t think the paper gives the amount of water used so it’s hard to know if the values are reasonable. It would have helped if a sample with an appropriate level of sodium chloride was employed to see how close these combination of additives were to the model containing sodium chloride- surely that would be the ideal and the correct comparison?

The Derringer’s desirability function has been calculated, but I think this seem the wrong way round. The guess is what ingredients are used and then the values for the dependent variables are given. Surely what is required is the desired parameters and then what is the best combination of ingredients, however to do that a salt control would be necessary.

I expect this is a piece of a wider set of work where product has been baked and the bake performance has been assessed and that would include sensory. This current work may hint at what was required, but I am sorry that as it stands I feel its data without a purpose.

Reviewer 2 Report

Unfortunately, this paper is not interesting for a broader public, as hardly anything can be learned from the mechanism behind, no idea of a reference dough compared to sodium chloride containing dough and too little information about the sample composition is given. Basically, the reader can only learn something when he/she wants to add one of the specific ingredients (which are unable to be identified based on the given description).

Some key aspects are listed below before a decent evaluation can be made.

Title:

It is more an evaluation than a report on the effect. Some additives do not replace the chloride, only the sodium. In fact, it is not a replacement as the sodium-containing bread is not tested. A title could be: ‘Evaluating the impact of different additives on rheological behavior of low sodium wheat flour dough’

Sea salt:

  • Please provide product name/number.
  • Line 65, should that be 21-27% and 31-35%?
  • About the composition: counting the maximal levels of each given component, we end up with 79 out of 100%. Please comment on the remaining 21% (or more).
  • The name sea salt is distracting. Regular sea salt might contain up to 98-99% of sodium chloride. The sample name should thus be changed to low sodium sea salt (the ‘sea’ can even be left out as that distracts from the fact it is containing low levels of sodium). Abbreviations should be changed accordingly.

Dry sourdough:

  • Please provide product name/number.
  • This samples needs more details. How does it occurs, is it a powder, lumps, … If a powder, how fine is the powder, as particle size might strongly impact the hydration properties.
  • Is there any residual microbial or yeast activity?
  • Some insights in the drying process (low or high temperature, freeze-drying, …)
  • When sourdough is added, why is there no compensation for the water content? Hydration will be strongly impacted, taking water from the remaining recipe and changing the rheological behavior.

Fermented sugar:

  • Please provide product name/number.
  • The abbreviation ‘SF’ is confusing, ‘FS’ would be more appropriate for a sample called ‘Fermented Sugar’. In fact, ‘fermented sugar’ is not a clear sample description itself, it is rather a ‘liquid extract of sugar fermentation’.
  • His appears to be a liquid. What is the water content?
  • Any ethanol inside?
  • Why are there residual sugars? And which sugars are those?
  • What were the starting sugars? How pure were they? What were the used fermentation strains?
  • Is there any residual microbial or yeast activity?

The evaluated wheat dough recipe:

  • Please provide a full ingredient list. It is not clear what ratios of flour, water and others are used. Which yeast is used and at what level? Any sugar in the dough?
  • The relevance of these results can not be evaluated, as the reader has no information on the effect of sodium chloride itself. What are the reference values for a salt containing dough? This seems a necessary addition to make this paper interesting for publication.

Optimal conditions:

  • Why are there different optimal conditions in the abstract and in the conclusion section?
  • Did you prepare doughs with the optimal dosages? Did they actually show the best results?

Significance:

  • Samples are evaluated as significantly different, but the p value is only below 0.1. It is not common to see such high p values to be evaluated as significant, in most case, 0.05 is the maximal value for p to be named significant. Is there a specific reason to use such high values and make it easier for samples to be significantly different?

Graphs:

  • Although efforts have been made to produce good graphs, they are not attractive and easy to interpret. It is a challenge for the authors to find more accessible ways and better visualization of their study.

Language and style:

  • Please check the lack of ‘spaces’ in the text, e.g. line 30, 161, 182, 209.
  • Please check verbs, e.g. line 55, 166, 168
  • Please check numbering of the equations.
  • Avoid the use of the same words or concepts too often. E.g. line 29-30 ‘globally’ and ‘in the world’ in one sentence is not needed, they point to the same thing…; ‘Nowadays’ in line 30 and 34.
  • A few examples of sentences to be improved:

Line 33-34: ‘… daily salt intake varies between 7 to 13 g which exceed in a quite large amount the WHO recommended doses’ should be adapted to ‘… daily salt intake varies between 7 to 13 g which largely exceeds the WHO recommended doses’

Line 34-35: ‘Therefore nowadays in many EU countries and not only, are being taken different measures to the governmental level in order to reduce the salt amount from the food products.’ Should be adapted to ‘Therefore in many EU countries and beyond, measures are being taken in order to reduce the salt content in food products.’

Reviewer 3 Report

The paper presents an experimental study about the effect of some ingredients of wheat flour dough (sea salt, dry sourdough and fermented sugar) on its technological/rheological properties. These ingredients should replace sodium chloride, the excessive intake of it could be dangerous for the human health. The paper includes a statistical analysis to highlight the relationships between independent and dependent variables of the dough, and the calculation of the desirability values. The topic and the results could be interesting for the journal and for readers, however the paper should be strongly improved.

First of all, the English language needs a very careful and deep revision: many grammar errors and typos (wrong characters, missing space between words, missing brackets, etc.) are present throughout the text.

The authors have recently published (in 2020) other two articles about very similar topics (references n. 20 and 37 in the present paper). The authors should clearly explain, in the “Introduction” section, the novelty of the present work with respect to the previous ones, in order to avoid possible confusion in the reader. Moreover, it would be helpful to better discuss, in the “Results” and “Conclusions” sections, about the main analogies and differences with respect to the outputs of the previous works.

The following major issues should be addressed by the authors.

  • In the abstract, when introducing the desirability value (line 23), a brief explanation of its meaning should be added, otherwise this information could be incomprehensible to most readers.
  • The authors should explain how the real and coded experimental values (Table 1) have been selected.
  • For each experimental condition, two runs were performed, and the average results were taken for the statistical processing. To give the standard deviation would help to evaluate the repeatability of the different measures.
  • The selection criterion of the model in eq. 1 should be explained.
  • The role of the residual (ε) in eq. 1 is not clear. Is it a constant? In this case it cannot be distinguished from . Is it different for each experimental point? In this case should be not present in eq. 1, if it is a previsional model. The authors should clarify this aspect in the text, or correct eq. 1 (just in case it is wrong).
  • At line 155 the authors report that “For the dough development time (DT) value no significant model was obtained.”. This statement should be quantitatively justified. The same for the statement at lines 206-207 “… whereas on gelatinization temperature (Tg, °C) no significant model was obtained.”.
  • At lines 208-209 the authors say that “The most significant model was those obtained for H’m where AdjustedR2 = 0.75fallowed by those obtained for Tmax where Adjusted R2 = 0.71.”. It is not clear the reason of this statement, considering that, in Table 2, there are three parameters (VT, VR, CR) with a higher Adjusted R2
  • The authors should explain what they mean with the expression “… due to its content…” at line 222, about the pH decrease due to the SD and SF addition to the dough.
  • The meaning of the sentence “However, the addition of SS… …of the PVmax” is not clear. In particular, it is not clear what is “SS in a linear form”.
  • The authors should explain how the desirability and the relative importance indices in eq. 2 were obtained or calculated.
  • A summary table with indication if each independent variable is significant or not for each dependent one would be very helpful.

Moreover, also the following minor issues are present and should be corrected.

  • The unit of measure of the proving time (lines 80 and 180) should be specified.
  • The numbering of equations 1 and 2 is duplicated.
  • The symbols in the first column of Table 2 and Table 3 should correspond to those used in eq. 1.
  • The development time abbreviation at line 155 (DT) is different from that given at line 78 (DDT).
  • At line 238, “… the linear terms SS, SD…” probably should be “… the linear terms related to SS, SD…”.

Reviewer 4 Report

Presented manuscript is well written and contains useful and practical  result of investigations. However, I have some comments and suggestions which must be taken into consideration:

line 32 - I suggest changing 2000 mg to 2 g, such units are used later in this paper

line 55 - please change the order of items 17 and 18 in the text and in the References chapter

lines 160, 195, 244 - I suggest adding the names of the Authors of the works that the Authors of the research refer to

line 286 - please remove the unnecessary parenthesis

line 372 - please insert spaces in the title of the journal

lines 378 and 380 - please delete the year of publication of the journal and place it after the title

line 399 - typing error in the title of the journal

Round 2

Reviewer 2 Report

I am very glad to see the major improvement of this manuscript. Surprised even to update and adapt in detail the samples, controls, significance and relevance perception. Well done!

Some minor text remarks:

23 The explanation of the number makes it a bit long for the abstract. Prehaps adapt this line to ‘addition with a decent desirability value (0.618 on 1.000).’ and then explain the scale later in the text. I would also at a statement that it was even better appreciated than the control bread with 1.5% NaCl. The latter is a very nice finding which was not included in the first version of the manuscript I believe.

82: ‘it may contain’ (no S)

89-91 The details on the microbial load can be left out. My question was aiming for residual yeast/fermentation activity but I believe that the provided information is fair now.

288 ‘followed’ with an ‘o’

305-307 please adapt to this version: ‘It is well known that through the fermentation process, the amount of lactic acid bacteria and produced acids increases which leads to a decrease of pH values’

399 ‘a significant’

402 ‘a more strengthening effect’

406 ‘due to its action’

408 ‘in its composition’

432 ‘it contains’

Reviewer 3 Report

The authors properly addressed the issues previously arisen and the paper quality was significantly improved.

In the reviewer opinion the paper is now suitable for publication after the few typos still present (e.g. “thefallowing” instead of “the following” at line 102, “anAmyloghraph” instead of  “an Amyloghraph” at line 108, “contribuite” instead of “contribute” at line 442) have been fixed.
